# Religion and Democracy in Argentina Religious Opposition to the Legalization of Abortion

**Marcos Carbonelli** [1,*] and **Maria Pilar García Bossio** [2]

1    Center for Labor Studies and Research (CEIL), National Council of Scientific and Technical
     Research (CONICET), Buenos Aires C1083ACA, Argentina
2    Institute of Social Sciences Research, Catholic University of Argentina (IICS), National Council of Scientific
     and Technical Research (CONICET), Buenos Aires C1107AFF, Argentina; mapilargarciabossio@gmail.com
*    Correspondence: mellimarcos@gmail.com

**Abstract:** This article analyzes the ways in which religious actors opposing the legalization of abortion adjusted their arguments and public actions to the Argentine democratic culture between 2018 and 2020. Data were collected through a qualitative research approach by conducting in-depth interviews with activists, studying public position statements in secondary sources, and analyzing pronouncements and interactions on social media platforms. Religious agents conceived of democracy as the rule of the majority that they intended to promote by means of secular arguments, demonstrations in public spaces, and the construction of electoral alternatives. Marginally, the categorization of feminism through conspiracy theories and the use of dilatory legal maneuvers ran counter to the logic of the expansion of rights. According to the empirical evidence gathered, the religious agents showed increasing adjustments to the language and criteria inherent to democratic life.

**Keywords:** religion; abortion; democracy; feminism; Argentina; qualitative research

## 1. Introduction

An indicator of the consolidation of democracy in Argentina has been the openness to diversity, the consolidation of the right to freedom of expression, faith, and autonomy over one's own life and body. One form of this openness has been the discussion about the expansion of so-called sexual and reproductive rights. The legalization of abortion in December 2020 was the latest milestone in a series of conquests in this field that started with the enactment of the civil divorce law in 1987, and then continued with the sex education (2006), same-sex marriage (2010), and gender-identity (2012) laws.

Similar to other parts of the world (Krannich 1980; Albarracín and Lemaitre Ripoll 2016; Campos Machado 2016; Graff and Korolczuk 2022), religious groups reacted against these initiatives, bringing them into question and mobilizing in order to uphold the legal protection of conservative moral orders. According to Casanova (1994), the defense of traditional life regimes is one of the reasons why religions step into the public arena.

The diversification of the conservative religious field (which in Argentina originally included only members of Catholicism before the arrival of evangelical groups in the first decade of the 21st century, see (Jones and Carbonelli 2012; Campos Machado 2018; Bárcenas Barajas 2022)) and, fundamentally, the fact that its actors regularly and systematically opposed any expansion of sexual and reproductive rights has motivated activists and their political allies to brand this coordinated religious action as "anti-rights". Such a label was rejected by conservative sectors, who differentiated between moral, economic, and social positions and contended that they were not overstepping the boundaries of democracy. In turn, this accusation of being "anti-rights" tended to unite, under a homogeneous conservative label, groups with very different positions in different areas, but whose common interest, particularly in the case we are analyzing in this paper, is the opposition to the legalization of abortion and other laws that threaten the traditional family model.

In this work, we examined this political statement and considered it as a scientific question: how does the public behavior of conservative religious agents depart from or conform to Argentine democratic culture? This question was relevant from a dual-theoretical perspective. On the one hand, as a Western country, Argentina participated in the global controversies between the religious identification of its citizens and the principle of secularism in the modern state, what Habermas (2006) called the constitutive tension between a secular state and a post-secular society.

On the other hand, after a century marked by military coups, the country's democracy faces the challenges of its consolidation stage, which are rooted in the two essential dimensions of this type of regime, as outlined by Dahl (1971): the opening of public debate and the free expression of ideas and opinions, and the participation and/or channeling of these ideas through the mechanism of political representation as deployed through party competition and voting.

Based on these considerations and on the conceptualization of democracy as a field of antagonisms where demands are channeled and disputes between opposing interests are settled (Laclau and Mouffe 2005), we defined democratic culture as the set of notions, knowledge, and representations that guide the competitive interaction of its participants and that permeates both the usage of the resources available through the democratic game and the respect for the rules of democracy and the outcomes of its conflictive dynamics.

In order to answer this question, we first analyzed what we called conservative religious groups' arguments and actions during the debate over the legalization of abortion between 2018 and 2020. Secondly, the elements considered in the analysis allowed us to identify the conceptions of democracy that informed conservative religious agents' public interventions and to what extent these were aligned with the above-mentioned normative principles.

It should be noted that in this paper, we considered conservative religious agents to be those who held conservative positions regarding the legalization of abortion, even when they did not share other ideological positions (e.g., on the direction of the economy, insecurity, etc.). However, it should be taken into account that some religious communities—though, certainly, a minority—advocated for the law with various levels of commitment and support. These included both historical Protestant churches and, for the first time, some Pentecostal communities, in line with what was happening in other countries in Latin America (De la Torre and Semán 2021). We also did not delve into the political affiliations of these actors since this issue transcends political parties and electoral fronts (and has even generated alliances between unlikely actors). We only mentioned it concerning the formation of political parties whose main identities were related to the rejection of the legalization of abortion.

The text is organized as follows: first, we provide an account of the controversy regarding the legalization of abortion in Argentina, focusing on the decisive debate that developed between 2018 and 2020, and that came to an end as the legalization bill was passed. Next, we present the religious arguments against legalization, how these positions were justified, and how conservative religious groups thought democracy should work. Afterwards, we examine some of their main actions, such as their mobilization in public spaces and the formation of political parties. We then explain the methodology used to collect the data, and finally, we provide our conclusions.

## 2. Materials and Methods

Methodologically, this work utilized a qualitative design. This allowed us to use an interpretive approach (Denzin and Lincoln 2012) to describe controversies, identify their milestones and most relevant actors, and understand their consequences in terms of political culture, as well as the implications of the latter for characterizing those who are religious.

First, we conducted an extensive literature review, after which we used secondary sources, both secular and those produced by religious news agencies. We examined online

periodicals, nationally circulated newspapers (*La Nación*, *Clarín*, *Página 12*, *La Izquierda diario*, *Infobae*, and *Letra P*), religious media (especially the Argentine Episcopal Conference and ACIERA's official channels), and related publications, such as press releases, proclamations, etc. issued during the period of the controversy (2018–2020)[1]. The information collected was systematized in databases.

This analysis was expanded with detailed research into online social media commentary, particularly on Facebook and Twitter, during the time when the controversies had developed. We recorded the milestones, actors, and languages involved. These data were processed by the COES Company through a social-listening system. The purpose was to analyze the stance of social media users on this controversy; identify the conversations for and against the issues, and the specific events that induced these conversations; detail the main arguments used by individuals for and against the bill; and identify the types of users that generated these conversations, as well as the most frequently used keywords and hashtags.

The analysis considered only public profiles in order to respect social media privacy policies. This limitation was applied specifically to Facebook, where most profiles created by users are private, whereas on Twitter, they tend to be public. Data were mined and then classified through artificial intelligence, using Metrix Bi, Sprout Social, Mention, and Power Bi tools. Human interpretation was subsequently applied in order to obtain more accurate parameters about the stances of individuals regarding the issues under study. The periods covered were from January 1st to December 31st in 2018, and from November to December 2020.

The keywords used for the search were (in Spanish): aborto, interrupción voluntaria del embarazo (IVE), pañuelos verdes, pañuelos celestes, #salvemoslasdosvidas, #lamayoríaceleste, #argentinaesprovida, #salvenalos2, #abortonoessalud, #abortoessalud, #niñasnomadres, #uniendovoces, #IVE, #abortosesionhistorica, #valetodavida, and #todavidavale[2]. It was thus found that during the periods covered, 287,029 posts and comments on this topic were generated by 80,649 authors. Out of these, 13% were media; of the rest, 65% identified themselves as female, and 50% were between the ages of 18 and 24.

July and August 2018, the period when the legal abortion bill was debated in Congress, were also the months during which most posts and comments had been made. Most conversations revolved around the bill (60.43%) and human rights (27.79%). With regard to religion, 35.43% of those who mentioned it used religion to justify their position against the voluntary termination of a pregnancy. It was found that, in social media, conversations in favor of legal abortion prevailed, with a 59% positive sentiment. Females (60%) and users between the ages of 18 and 35 (67.54%) generated most of the content in support of the bill.

These data allowed us to observe a group that played an important role in the legalization campaigns in the digital world: young females. These data from social networks complemented what had been mentioned by several people in other cases: The possibility of legalization went hand in hand with the "daughters' revolution". At the same time, the strong presence of pro-choice positions in social networks showed that this had been another area of contention that, to some extent, mirrored what was happening in the streets and in Congress. These data also allowed us to question an explanation presented by both sides of the conflict, wherein pro-life people seemed to be the most active in the virtual world (taking into account the methodological caveats mentioned previously).

In parallel (between December and March 2022), we conducted 11 in-depth interviews (Arfuch 1995) of key actors, including males and females, who participated in these controversies both from conservative religious groups and organizations or movements that sought to ensure the expansion of these rights. Given the COVID-19 pandemic restrictions, many interviews were carried out using Meet and Zoom, while at all times following present-day social science guidelines on research ethics (Meo 2009). The interviewees were five evangelical leaders (in one case, two people were interviewed simultaneously) and two pro-life Catholic priests. Among those who advocated for legal abortion, we interviewed two feminist activists, a Pentecostal female pastor, a leader from Catholics

for the Right to Choose, and a top official of the federal Executive Branch. We sought to highlight a wide range of positions and activism within each group, choosing people with a high public impact on the debate. Interviews were transcribed and processed with the Atlas.ti software (Chernobilsky 2006), according to the Grounded Theory methodological guidelines (Soneira 2006).

In this way, the information gathered from traditional media (newspapers), social networks, and in-depth interviews allowed us to reconstruct the arguments and strategies used by both conservative and feminist groups for the different areas of conflict over the legalization of abortion.

## 3. Results and Discussion

Although the controversy about the legalization of abortion reached a peak during the Congress debates that were conducted between 2018 and 2020, it had started much earlier, during the so-called "second wave" of feminism in the 1960s and 1970s (Tarducci 2018). At that time, the goal was to expand the right to abortion beyond the two situations in which it had been allowed by the Criminal Code since 1921: if there was a danger to the mother's life or health that could not be avoided by other means, and if the pregnancy was the result of the rape of a woman with mental illness (this last case would also include any female victim of rape, according to the Argentine Supreme Court's 2012 F.A.L. ruling[3]) (Esquivel 2014; García Bossio 2019). After the restoration of democracy in 1983, the demand for legalization increased, and in 1988, the Commission for the Right to Abortion (*Comisión por el Derecho al Aborto*) was created. In 1992, the first-drafted bill about contraception and abortion was submitted to Congress. However, at the time, the possibility of debating it was thwarted by the strong rejection from various social sectors and the Catholic Church. Simultaneously, civil society organizations that rated themselves as pro-life, and where both religious and non-religious agents opposing legalization converged, were being organized (Morán Faúndes 2015).

In 1999, the Coordination Unit for the Right to Abortion (*Coordinadora por el Derecho al Aborto*) was created and recruited feminist and political organizations, unions, and the Catholics for the Right to Choose group (*Católicas por el Derecho a Decidir*) (Alanís 2005). At the 2003 National Women's Meeting, this group proposed the use of a distinctive green headscarf as a symbol of life and hope (Felitti and Ramírez Morales 2020). The choice of this triangle-shaped headscarf entailed adopting and giving a new meaning to the one used by the Mothers of Plaza de Mayo (the leading human-rights activists in the country) as a political device. Quintana and Barros (2020) have rightly pointed out that, in the Argentine political tradition, a headscarf is an instrument of denunciation and remembrance, and an icon indicating that the person who wears it is part of a collective project.

In 2005, the National Campaign for the Right to Legal, Safe, and Free Abortion was launched, with the participation of NGOs, female groups, and social organizations around the entire country. From then onwards, bills to decriminalize and legalize abortion were submitted in 2007, 2010, 2012, and 2014 (Tarducci 2018), but none of them were discussed. It was only in 2018 that a new bill was debated and passed the House of Representatives with 131 favorable votes and 123 against it, though it was rejected by the Senate, with 38 against 31 votes. As we later explain, the bill's failure after this first congressional debate was due to the strong presence of conservative religious groups, mostly Catholics and Pentecostal evangelicals, who used different arguments and performed various actions in order to increase the pressure on legislators (Gudiño Bessone 2022).

A new, amended bill was later submitted, this time with the explicit support of the President of Argentina, Alberto Fernández, and was passed by the House of Representatives with 131 affirmative and 117 negative votes, and with 38 positive and 29 negative votes in the Senate. The voluntary termination of pregnancy was thus legalized until the 14th week of pregnancy, based on the will of the pregnant person, and also after that period in cases of rape or danger to a pregnant person's health or life.

This feminist victory was fueled by the convergence of mobilization cycles, political decisions, and efficacious activist performances. From a mid-term perspective, the fact that the Comprehensive Sexuality Education (2006), same-sex marriage (2010), and gender-identity laws (2012) had previously been passed (Jones et al. 2010, 2014; Vaggione 2010, 2013; Jones and Dulbecco 2015) paved the way for the legalization of abortion, in that their enactment evinced the weaknesses of the opposing religious power and the greater receptivity of the political class to the demands of sexual minorities (Esquivel 2015). This process was reinforced by the emergence of the *Ni una menos* (Not One Less) movement in 2015, which facilitated the entry of younger generations of advocates into the ranks of feminist activism, and placed the demand for body autonomy on the agenda, thereby taking the debate about abortion to a society-wide level (Rebón and Gamallo 2021). Although the incumbent president Alberto Fernández had already voiced support for this cause during the 2019 electoral campaign, his government decided that the bill should be pushed by the Executive Branch due to its ideological conviction and because this would give it an opportunity to regain leadership and agenda-setting power in the context of the existing adverse political conditions, marked by the pandemic crisis and the need to address the foreign debt issue with the IMF. Finally, the analysis should not overlook the effectiveness of the feminist movement's campaign, which was able to sensitize society at large and bridge partisan divides, garnering the support of stakeholders from across the entire political spectrum, as well as artists and communication leaders, among others (Borda and Spataro 2018).

After this overview, the following section considers the religious arguments against legalization, in order to analyze to what extent they have adjusted to Argentina's present-day democratic culture.

### 3.1. Democracy-Conforming Arguments against the Legalization of Abortion

In the case of Argentina, the set of arguments and actions deployed by religious agents against the possibility of abortion becoming legal has constituted what may be called a symmetrical antagonistic space. The notion of symmetry refers to these agents' decision to mirror each of the arguments and actions of the feminist movement and its allies. In this regard, we agreed with Rebón and Gamallo (2021) in characterizing this opposition as a counter-movement, i.e., an organized social group that is set up as a response against the emergence of a collective subject that pre-exists the group and acts as its adversary in a democratic arena.

Conservative religious agents have sought to disprove, in particular, two of the feminist movement's central arguments. The first was the view of abortion as a severe public health issue based on the figures of mortality from clandestine abortions, which affect mostly low-income females. This is the perspective that has historically informed the National Campaign for the Right to Legal, Safe, and Free Abortion, and which was summarized by a well-known slogan: 'Sexuality education for deciding, contraceptives for avoiding abortion, legal abortion for avoiding death'. The second argument, derived from the liberal worldview, asserted women's sovereignty over their own bodies. In this regard, the slogan was 'my body, my decision', and its goal was to question the patriarchal regime that controlled female bodies, as it rejected the idea of pleasure and individual autonomy over a person's own life plans.

To counter these points of view, the religious opposition brought a number of arguments into the public arena, most of which had been deliberately developed without resorting to the logic of religious discourse and its justifications based on dogma and Biblical principles. This form of public religion was termed 'strategic secularism' by Vaggione (2005): this concept alludes to these agents' decision to adapt their discourse to the demands and modalities of modern democratic times in order to make their position more effective and compelling. This practice already had an important previous history, in particular, the interventions of religious actors against the expansion of sexual and reproductive rights, especially those afforded by the Comprehensive Sexuality Education

(2006) and same-sex marriage (2010) laws. As shown by a great deal of research (Jones et al. 2010, 2014; Carbonelli et al. 2011; Felitti and Prieto 2018; Torres 2018), such interventions have set a precedent for the use of discourse based on extra-religious foundations, since philosophical, legal, and bioethical reasons have been privileged over the direct use of Biblical justifications.

Examples of arguments designed according to the matrix of strategic secularism included the efforts to refute the figures of female mortality from abortion malpractice and, fundamentally, the bioethical–philosophical debate about the subject status of the fetus (Irrazábal 2022). Using arguments from current bioethicist-related thought, conservative religious groups upheld that human life began at fertilization, and that this life had rights and involved a distinctive subject who was different from the pregnant woman.

> So, when the debate was raised and started to develop, we as Christians stood up in defence of these concepts: the concept of the life in the womb that you cannot kill or interfere with, and that a woman has no right to do that, either. That is, because it is a new life, so when these two cells come together and start to form a new being that is genotypically different from his or her parents, we understand, on the basis of this definition, that we cannot accept this as a right in our society or our country because it is not, because the woman's right is being prioritized over the fetus's and the embryo's right ( . . . ) I have been told: 'But you are a Christian, you have neither voice nor authority to defend life'. So I say, no, that's not right: on the one hand, what I say comes from what I feel and from the word of God, and on the other, it comes from science, I cannot ignore what science says, you cannot ignore that this is an unborn human being, so if you want to make one decision or other, that's all right and that's respectable, but we understand, based on this concept, that we cannot allow that. (Director of the bioethics department of the Christian Alliance of Evangelical Churches of the Argentine Republic—ACIERA, by its Spanish acronym; personal interview, 24 February 2022)

This was the origin of the 'let's save both lives' slogan, which, on the one hand, claimed that a pregnancy involved two human beings and, on the other, conveyed a concern about the fate of those who face an unwanted pregnancy. On top of this 'defense of both lives', as it was termed, legal arguments were presented. Those who opposed abortion in the 2018–2021 debates stressed that the Argentine state was a signatory of international treaties with constitutional status, and that several of these, such as the American Convention on Human Rights, enshrined the protection of human life by the state 'from the moment of conception'. Under this logic, the voluntary termination of pregnancy bill (IVE, by its Spanish acronym) was unconstitutional. This was in line with what Vaggione (2021) called reactive juridification: the attempt to make one's own values fit in within legal frameworks in order to ensure that one's own position could be universally applicable.

This perspective was aligned with a worldwide outlook and with the defense of values considered to be part of the Argentine nation's cultural heritage. Whereas supporters of the voluntary termination of pregnancy pointed out that most countries of the so-called developed world legalized abortion decades ago, those who opposed it had used the same argument to uphold that the intention to decriminalize abortion as part of a global strategy with a eugenic intention and with the purpose of imposing birth control on peoples in developing countries. In the words of a 'shanty-town priest'—a Catholic priest belonging to the Option for the Poor group who works with the most disadvantaged social groups in the greater Buenos Aires area[4]:

> We even showed surveys about this issue carried out in shanty towns, and they couldn't care less, so I don't think there was a debate, it was rather something that had to be imposed, which involves reducing the Argentine population and well, many other criteria that lead to the same. One is abortion, another one is fostering childless couples, in many different ways and, well, thousands of

other forms. I think this is an imposition, evidently from abroad. Rockefeller had already started to say this in his time. This was upheld, during all this time, with the support of the loan and credit organizations, of the IMF, and it's no coincidence, it's no coincidence that [the then president] Macri, in 2018, at the time of the IMF visit, should raise the issue of abortion; it's no coincidence that Alberto Fernández, now that he has to negotiate with the IMF, should also raise this issue, so I think it is Argentine intellectuals that are the most contradictory, rather than the poorest people. (Father Pepe Di Paola, personal interview, 25 March 2022)

This last point was interesting, because it showcased the symmetry pointed out previously. Both the feminist movement and the conservative religious opposition portrayed the actions of their adversaries as part of a global conspiracy. Therefore, feminist circles often referred to the local Catholic Church's connections with the Vatican's directives and, more recently, to the growth of evangelical churches as part of an international scheme orchestrated by the international right to stop the 'green tide' (as the movement for legalization calls itself, making reference to the color of its distinctive headscarves, Bianciotti 2021). Conversely, opposition groups have often pointed to the alleged financial support received by feminist movements from international NGOs and from European and North American states as confirming the global birth-control theory.

The counterarguments used by the movement against the legalization of abortion have included different types of challenges to representative formats. On the one hand, they have criticized the feminist movement's claim that they speak on behalf of female people as a whole, and in particular, poor female people. Availing themselves of the language of social science, they have questioned the figures of female people who had died due to clandestine abortions, disseminated by feminist organizations and even the National Ministry of Health, and provided instead their own alternative surveys.

In addition, they have strongly criticized the formal representation of the legislators who had the power to decide whether to pass the bill. Different leaders of the opposition movement have recommended that lawmakers should listen to the voice of the people and the will of the majorities, who were supposedly against the legalization of abortion. This was observed in both Catholic and evangelical leaders:

So we think that a democratic mistake was made in this case. Because, you see, the fact that you represent the people does not mean that you can vote according to your own ideology, your own way of thinking; we expected them to cast their vote according to the majority's way of thinking, in a country where the Constitution is in favour of both lives. This is our interpretation of how democracy went astray. (Executive Director of ACIERA, personal interview, 24 February 2002)

I think that, deep down, they have links with an international proposal. Argentina is not Buenos Aires ( . . . ) they think differently from the rest of the country, so in a shanty town you will find that practically 100% of people were against abortion. Some of them even told me that they had been to the demonstrations because they had been taken to them by organizations, and they would go there with a light-blue headscarf and were made to take it off because, you see, they went to the square [Plaza de Mayo], I didn't go there, but they went to the square and stood on one side. These things were the consequence of Argentine politics, I don't think this just happened innocently. (Father Pepe Di Paola, personal interview, 25 March 2022)

Conservative religious actors have focused particularly on those legislators who represented the historically more conservative provinces of the Argentine hinterlands and accused them of ignoring the voice of the population in the provinces. They thus created a rhetorical fracture between a cosmopolitan center, influenced by the ideas of countries that

subscribe to a certain sexual and reproductive rights agenda, as opposed to the hinterland provinces, portrayed as guarantors of national values.

Analytically, the importance of the arguments against the legalization of abortion was that they provided a glimpse into conservative religious groups' conceptions of democracy. Such worldviews have fluctuated between two different but not incompatible paradigms. On the one hand, the appeal to secular arguments evidences an acknowledgment of contemporary democracies' liberal matrix, where the goal is to convince different audiences through sound arguments that may be detached from particular interests, so that they can be presented as universal. The idea of defending the rights of the embryo or unborn baby also fits with the liberal matrix of democracies, because it rests on their intrinsic defense and safeguarding of individual guarantees.

On the other hand, challenges to formal representation and references to fictional majorities ignored by the political dynamics imply a different paradigm, according to which democracy is the rule of the majority, whose will can be intuited and perceived in public demonstrations, but fundamentally, through first-hand knowledge of popular sectors' day-to-day life and of the values that inform social interactions in their neighborhoods.

### 3.2. Democracy-Conforming Actions

The arguments reviewed were coupled with a repertoire of collective action (Tilly 2009; Tarrow 2012), which also employed a symmetrical approach. The feminist movement had performed various actions in diverse public arenas, such as in the streets, on social media, in legislative committees, and in the hallways of Congress. At each of these places, it made sure to present a unified discourse and demand and to cross over the boundaries and barriers of party structures by permeating their grassroots and avoiding the usual partisan cleavages in order to propose another one: legal abortion, yes or no. A noteworthy aspect of the feminist collectives' public performances was that the green headscarf, as previously mentioned, was rendered increasingly visible. It had been part of the campaign since the early 2000s, but now it was displayed en masse. Advocates started to wear it around their wrists, ankles, and heads, as well as on their bags, thus creating a visual effect in the urban landscape. The view a crowd of green headscarves in public life had the effect of showing a 'green tide' of demands that were integrated into everyday life (and which also populated social media images and imagery in the form of stickers). The symbolic bridge that linked the Mothers of Plaza de Mayo's original headscarves with the current scarves was that, in both struggles, it was female people that played the major role (Felitti and Ramírez Morales 2020).

Faced with this public action, religious opponents opted for a plan that consisted of occupying the same spaces, holding debates in the same public arenas and imitating the modus operandi of the massification of individual supporters by using headscarves as a political technology. The 'light-blue ones' (who sought to associate their movement with the nation, since light-blue is one of the color of the Argentine flag) also organized demonstrations in public space, where they held banners displaying slogans such as 'let's save both lives' and 'all lives are valuable', and they proposed the use of light-blue headscarves, thus turning this color into the hallmark of the pro-life movement. They countered social media hashtags such as "#Abortolegalya" and "#Seráley" ('legal abortion now' and 'the bill will be passed') with '#salvemoslasdosvidas', "#todavidavale", and '#noseráley' ('let's save both lives', 'all lives are valuable', and 'the bill will not be passed') (Calvo et al. 2021), so as not to give their opponents any advantage or let them prevail in any field, whether physical or virtual.

In addition to reacting to the feminist movement's initiatives, these religious actions also followed the traditional practices of Argentine movements (Pérez and Natalucci 2010). The country has a profuse history of the mobilizations of different ideological signs and affiliations, and of advocates who have taken their causes to the streets as their theater of operations. Religious groups have not been an exception to this rule, which, as pointed out by Fillieule and Tartakowsky (2015), has also guided the repertoires of collective action

in other parts of the world, where democratic life allowed the use of public space as a territory for mobilization and for raising awareness about certain issues. Ultimately, conservative religious groups' political actions were in line with the regular patterns of citizen participation in contemporary democracies.

We believe it was relevant to focus on three specific actions of the repertoire analyzed: shaming, court filings, and the creation of political parties. We paid particular attention to shaming because public harassment involves placing negative pressure on political decision-makers, something that ran counter to open argumentative exchange and the guarantees of reflexivity. It should be noted that this method was also used by some sectors of the feminist movement (Laudano et al. 2020)[5]. The light-blue sectors identified those legislators in social media who were 'between two minds' at the moment of casting their vote and even appeared on their doorsteps. Feminist leaders were also identified and harassed. In the context of the 2019 general election, the conservative sectors demanded that candidates should openly state their position in the debate. In the case of the feminist movement, some key pro-life actors were also identified, and a directory with their names, positions, and addresses was even posted on social media, though it was later deleted after a brief controversy.

Regarding the court filings, we should underscore the submission of the *amparo*[6] filings by light-blue organizations in order to avoid non-punishable abortions (as defined by the regulations in force before the law was passed) and to prevent the actual law from being enforced upon its enactment. It should be pointed out that the legal arena was a field of dispute before, during, and after the controversy, being a form of 'politics through other means' (Smulovitz 2008), as while legal actions were now being taken to curb the expansion of rights, in the past, they had been used as a resource to broaden them (Fernández Vázquez 2022). Before the IVE, the early disputes about the scope of the law had been held in the context of federal and provincial court cases (such as the F.A.L. case mentioned above).

Finally, regarding the creation of light-blue political parties, two were formed, each with its own distinctive characteristics—among other, smaller-scale experiences—and majority parties also appointed some light-blue candidates to run for legislators (Semán and García Bossio 2021). In 2019, one year after the Senate rejected the decriminalization bill, the NOS party competed in the primary elections[7], with Juan José Gómez Centurión running for president and Cynthia Hotton, for vice president. The former was a Malvinas war veteran who had close relationships in Catholic and military circles in Argentina and whose public speeches extolled family values and national sovereignty. Hotton, on the other hand, had a longer political career, since she had served as a national representative for the center-right alliance between the RECREAR (Recreation for Growth) and PRO (Republican Proposal) parties, and was more firmly established within the religious space, being part of a well-known family lineage in the evangelical world. During her term as a legislator, she stressed her religious identity, becoming one of the top leaders of the opposition to the same-sex marriage law and creating a political space, *Valores para mi País* (Values for my Country), where she tried to secure the monotheistic religious vote through an agenda based on values such as the defense of the traditional family, the opposition to the legalization of abortion, and her criticism of the political class's corrupt practices (Carbonelli 2020).

During its campaign, the NOS party merged nationalistic slogans and demands (the defense of national sovereignty) with elements that came from pro-life organizations. Opposition to the legalization of abortion and to the teaching of sexuality education with a 'gender ideology' in schools were prominent in its electoral campaign. Although the party managed to obtain more than the required number of votes in the open primaries and compete during the general election, it did not have the capacity to change the political dynamics, with the electorate becoming polarized between the two majority political alternatives, and disbanded upon being defeated.

The second light-blue political experience was connected to the first one: Cynthia Hotton did not give up on her political ambitions and ran again as a national representative

of the Province of Buenos Aires in the 2021 mid-term legislative elections. This time, the second candidate on the list was Gastón Bruno. Both of them constituted the *Más Valores* (More Values) party. Here, the campaign underscored again the protection of the Christian conception of the family, and although abortion had already been legalized, the party highlighted its light-blue identity that had been developed during the controversy and promoted a feeling of discontent with the traditional political parties that had made legalization possible.

Bruno was also firmly rooted in the evangelical world. He served as vice president of the Christian Alliance of Evangelical Churches of the Argentine Republic (ACIERA) and held management positions in the Department of Culture and Education of the Province of Buenos Aires. This represented his political space *Gobernar Bien* ('Good Government') and as an ally of *Cambiemos* ('Let's change'), a center-right electoral front that included PRO, the Radical Civic Union, and other political parties. The *Más Valores* party was not any more successful than its predecessor, NOS: Although it managed to muster the required number of votes in the open primaries (after filing a court claim to have the vote count reviewed), it did not win a single seat in the general election.

Ultimately, conservative religious groups generally followed the classical repertoire of actions adopted for participating in the public sphere, such as taking to the streets, conducting debates on social media, and writing opinion articles in newspapers. Their most radical actions were the *amparo* court filings and public shaming. However, they were not the only entities to use these methods, since feminist groups had also used them in the past and continue to do. Regarding the strategy of creating political parties, its failure showed the impossibility of reducing the Argentine party system to religious options (Prieto 2014; Carbonelli 2018b), as well as the effectiveness of the feminist movement's strategy, targeted at making its proposal more pervasive by bridging electoral cleavages.

## 4. Conclusions

The analysis of the controversy that developed in Argentina, except for very few and minor exceptions (such as public shaming and the use of insults in social media exchanges) allowed us to show that conservative religious groups adjusted their arguments and repertoires of collective action to the customs, practices, and rules of democratic life. Therefore, they have participated in the formal debates in the committees of both Houses of Congress in Argentina and mobilized in different public arenas, such as in the streets, mass media, and social media, according to the traditional patterns of actions adopted by social movements within their society.

Recognition of this political game arena by conservative religious groups has been partly evidenced by the conceptions of democracy that they have showcased in their public interventions. Rhetoric mentions of a majority opinion that was not efficiently represented were conjoined with the appropriation of the language of human rights and with positions clearly inspired by liberal thought. A careful scrutiny thus revealed that the arguments swung between the allusion to moral majorities and the freedom of conscience as the ultimate guarantee of rights. These dynamics were also apparent in other social movements with an active participation in the public sphere. Furthermore, the demands of the feminist movement also combined these two lines of thought: both the liberal perspective that underlies a claim such as 'my body, my decision' and the majority argument implicit in demanding the legalization of abortion on the grounds that it was supported by a pre-existing social consensus.

Such adjustments by religious actors were no coincidence. Their regularity exhibited a consolidated and global modus operandi (Vaggione 2021). We can now to go back to, and problematize Vaggione's concept of strategic secularism. Was this adjustment to democratic dynamics merely pragmatic, or did it evidence a genuine assimilation to the rules of democracy? Was the public intervention of religious actors a problem for diversity in democracy, as that would imply an imposition of their own beliefs as those to be followed by the whole society, or was their intervention a reasonable part of the public debate and a

genuine intention to offer public reasons/justifications understandable to all socio-political actors? Considering the constant use of resources and actions that conformed to present-day democratic formats, the acceptance of the results of the controversies in which they had been involved (i.e., the legalization of abortion in Argentina) and, fundamentally, the absence of challenges to the democratic system per se, we were inclined to speculate that the second alternative was true for both questions.

In other words, the evidence collected in our research and summarized in the previous sections allowed us to claim that strategic secularism was also normative secularism: The stability of democracy as a government regime and a form of social life in the country gave rise to languages, procedures, and requirements that religious agents acknowledged as legitimate when expressing their demands and opinions in public spaces. Having fortunately become permanent, the democratic culture established parameters for participation that were actively respected by religious groups (Carbonelli 2018a).

Far from being an inconsistency or a danger, the fact that religious groups adapted to different argumentative formats confronted democracy with a test and a reason for reflection. It forced its actors and participants to consider the foundations that regulate our living together more thoroughly. In other words, that conservative religious groups should avail themselves of the human-rights paradigm to hinder the expansion of rights in the sphere of intimacy could appear paradoxical; however, we believe that this also raised the bar of the debate about the present-day validity of the principles of democracy.

At this point, we agreed with Vaggione's claim that religious politicization, even of conservative agents, could have a positive outcome: Not only did this phenomenon channel the expression of disagreement with the transformation of a social order, but it also legitimated the inevitably conflictive nature of democratic life while posing new challenges and questions. In our opinion, the analyzed case also confirmed Habermas' (2006) thesis of mutual and complementary learning between religious and non-religious actors in contemporary democratic societies. While the former incorporated the formal and informal rules of legal competition into their political actions, the latter were initiated into the recognition of cultural diversity and the practices of tolerance.

In this view of the relationship (virtuous or not) between religious agency and democratic life, conservative political parties were especially worthy of attention. The parties created by morally conservative Argentine actors obtained only meager results, thereby confirming a historical trend, i.e., that religious affiliation could not be translated into electoral behavior. This allowed for the comparison with other Latin American cases, in which morally conservative parties had formed as an attempt to influence politics.

As a closing remark, we would like to mention two findings that the data collected so far helped us recognize and which constitute fertile research avenues in the fields of study about politics and religion. First, our attention was drawn to the prominence that the judiciary has gained over the last few years, as a point of convergence of the demands that intended to challenge the established social order. The litigation to counter the extension of sexual and reproductive rights took the form of both amparo filings and conscientious objections (Bosio et al. 2018; Puga and Vaggione 2018). It is imperative for sociology to analyze the fabric of these demands (the contacts between judges and conservative groups and the elective affinities between the judiciary and different social groups) and for political philosophy to delve into the challenges that such interactions could pose to the future of democracy.

Secondly, the power of conspiracy theories in public debate cannot be overlooked. This article referenced the characterization of feminism and sexual diversity movements as appendages of overall birth-control strategies allegedly orchestrated by central countries and international organizations. Similarly, we mentioned the concern voiced by progressive sectors in the region about the conservative advances in different countries, considered as an epiphenomenon of a plan organized by US churches. Upon analyzing the usage of these discourses, it is necessary to ascertain their actual influence on public debate, focusing

on how widespread they have become among citizens, as a whole, and distinct from the intense minorities who habitually employ them.

**Author Contributions:** M.C. conducted interviews, processed the data, and wrote the analysis in this article. M.P.G.B. conducted interviews, processed the data, and wrote the analysis in this article. All authors have read and agreed to the published version of the manuscript.

**Funding:** This research was funded by the Latin American Council of Social Sciences (CLACSO), grant program "Threats and Challenges to Latin American Democracies: Rights under Question?". July 2021/April 2022.

**Institutional Review Board Statement:** Not applicable.

**Informed Consent Statement:** Informed consent was obtained from all subjects involved in the study.

**Data Availability Statement:** The data presented in this study are available on request from the corresponding author. The data are not available to the public due to commitments to CLACSO, the institution that financed this research.

**Acknowledgments:** We are grateful to the people we interviewed for their generosity with their time and perspectives. We would like to thank CLACSO for funding this research and for providing the space to share our ideas. We are especially grateful to Eduardo Rinesi for his mentorship on the project; and to the research team in Costa Rica (Andrey Pineda Sancho and Arantxa León Carvajal) for the exchange of ideas.

**Conflicts of Interest:** The authors declare no conflict of interest.

## Notes

1 Particularly important were the interventions of conservative religious groups and feminist activists in the consultative commissions that preceded the parliamentary debates in both the House of Representatives and the Senate.

2 Translator's note: The keywords were abortion, voluntary termination of pregnancy (IVE), green headscarves, and light-blue headscarves; the hashtags indicated 'let's save both lives', 'the light-blue majority', 'Argentina is pro-life', 'save the 2 of them', 'abortion is not health', 'abortion is health', 'girls not mothers', 'joining voices', 'IVE', 'abortion historic session', and 'all lives are valuable'.

3 The F.A.L. ruling was a decision by the Argentine Supreme Court that settled the dispute about the interpretation of Article 88 of the Argentine Criminal Code: Before abortion was legalized in 2020, some actors of the judiciary and civil society could claim that abortion was not subjected to punishment if the person had become pregnant as a consequence of rape, whereas their opponents upheld a restrictive reading of this article, according to which abortion was not punishable by law only if rape had been committed against a 'feeble-minded or insane' person. The Court finally ruled in favor of the first group after addressing a controversial case in the province of Chubut, where a young person had become pregnant after being raped by their stepfather.

4 In Argentina, religious organizations often used the social work that they conduct in the most marginalized areas of society, supplementing public policies as a kind of 'spare tire', to gain legitimacy before the state and to claim recognition as public representatives of popular sectors (Carbonelli 2015).

5 In Argentina, shaming originated in a different context, as it had been associated with the fight for justice by the organization *Hijos e Hijas por la Identidad y la Justicia contra el Olvido y el Silencio* (HIJOS) (Sons and Daughters for Identity and Justice against Forgetting and Silence) during the 1990s, as a form of social condemnation of the perpetrators of repression during the last civic–military dictatorship who had not been criminally prosecuted (Cueto Rúa 2010). In this regard, interestingly, a strategy used by a sector that fought for human rights to make injustice visible was given a new meaning by both feminist groups and conservative sectors. Whereas this practice was adopted by more conservative sectors for the first time, the extent of its use by feminist groups deserved reflection, especially if conceived of as a strategy that could be complementary to, but never fully replace, the demand for the proper functioning of institutional channels (Di Corletto 2019).

6 An *amparo* action is a legal measure that can be admitted against any action or omission by a public authority that, in a manifestly arbitrary or illegal manner, causes actual or imminent damage, restriction, or alteration of, or threat to, rights or guarantees that are explicitly or implicitly recognized by the National Constitution, except individual liberty, which is protected by *habeas corpus* (Law N° 16.986/1966).

7 In Argentina, before general elections, open, simultaneous, and mandatory primary elections (PASO, by their acronym in Spanish) are held. They establish the minimum number of votes required for a candidate to run for office in the general election while also making it possible to settle internal party competitions with the participation of the citizens as a whole. Voting is compulsory for all Argentine citizens over the age of 18 and optional for those over the age of 16.

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
