# Peer review of "Religion and Democracy in Argentina Religious Opposition to the Legalization of Abortion"

_religions, doi:10.3390/rel14050563_

Round 1

Reviewer 1 Report

The paper does a very interesting analysis of the debate on the legalization of abortion in Argentina, and I find it very promising. However, I think it needs some more work in order to be publishable.

First of all, in political theory terms, in my opinion the authors mistake democracy with liberalism. They talk about religious actors' adaptation to Argentina's democratic culture. However, in my opinion, what they describe is indeed an adaptation to the liberal democratic discourse, carried out by translating their religious arguments in more liberal terms. This is a phenomenon that was well described for example already in the 1990s by Matthew Moen about the Christian right in the US. Moreover, I think the authors should also mention the theoretical debate about the acceptability of religious arguments in the public sphere, or the need to translate them, between Rawls and Habermas.

The idea of 'adaptation' itself at times also looked, in my opinion, a little biased, as if the authors implied that the right, democratic point of view is that of the pro-abortion militants. Although I tend to agree with them, they must be aware that this is not the only point of view. In my opinion, a more neutral attitude is recommended (in some parts of the paper this is not a problem, because there is already a good balance, but some parts can be improved).

In terms of local political culture, I also wonder why the authors never mention the peculiar political culture of the country which is an heritage of peronism and its idea of organic community. Maybe, this is not relevant for the paper, but I think it could be interesting considering it.

In structural terms, I don't understand why the methodological paragraph is towards the end. It usually finds its place after the introduction and before the analysis, since it helps the reader to better understand this latter.

I also think that the authors underused the interviews. In the paper there are just a few long quotations from them. I recommend a more extensive use of this material, to support the authors' points throughout the paper.

Author Response

Thank you very much for your comments, they were very useful for the revision of the article. We have corrected the sentences that implied a position on our part and clarified the proposed points.
We modified the position of the methodological section (we had placed it there following the article model suggested by the journal). The interviews were central material for our analysis, but due to the proposed length of the article, we were only able to make some direct quotations. We hope to be able to continue working with them in the future.

Reviewer 2 Report

The manuscript puts forth three interesting points, one empirical, one conceptual and one theoretical, that is, respectively: Argentinean conservative religious groups’ strategic alignment with “democratic life”, the “normative secularism” implied in such alignment and the theoretical challenge such normative secularism by/from conservative religious groups and their agendas brings to “the principles of democracy”

What concerns me is the manuscript’s methodological bases:

Though there is a “materials and method” section where (some of) the religious agents that were approached as research subjects are described (p. 9), it is unclear what the author/s mean by “conservative religious groups” throughout the manuscript. Other than ACIERA (p. 4, 5), the Catholic group related to option-for-the-poor movement (p. 5) and the Argentinean Episcopal Conf. (p.8), what other religious organizations or groups were researched? More importantly, how/why were those organizations chosen as subjects? The set of those “conservative religious groups” and whatever (soft) sampling criteria was used for their selection should be spelled out, especially for the audiences that are not familiar with the Argentinean Christian field.

Who exactly are the “conservative religious group/s” that carried out the shamings and the amparos (p. 6-7) that are described as “democracy-conforming actions”? This should be made explicit as well.

Why the interview quotes come only from two religious organizations/sectors (ACIERA and option-for-the-poor group)? Are they representative of the Catholic Church and the universe of Evangelical/Pentecostal congregations in Argentina? What sectors within those two fields would they represent? This should be spelled out as well, together with an expansion of Note 2 and, preferably, a relocation of the expanded note within the main text.

Why do the “materials and method” section contain data (ls. 428-438) that are not further analyzed in the “results and discussion” section? This should be spelled out, or the data has to be either removed or relocated and expanded on in the text.

Why do the “materials and method” section is inserted after the results. I am OK with flexibility for traditional writing formats but, in this case, I find the location of this section (just before the conclusions) an additional issue that complicates the reading of the manuscript.

Should not the link in p. 11 direct readers to a database or other collected data? A link to the company that was hired seems pointless to me.

In general, the sweeping statements throughout the manuscript about “conservative religious groups” and “democratic life” in Argentina might match the comprehensive analyses that are described in lines 401-423 and 439-450, yet do not seem to match the low number of data sources (or religious organizations) actually quoted in the text.

In my opinion, a more transparent and focused account of the research methodology has to be presented for the manuscript to be publishable.

Author Response

Thank you for your comments, which were central to the revision of the article. We have clarified the use of the term "conservative religious groups" in the text, appealing to a loose description that allows us to include religious actors who may have different views on other agendas, but who oppose abortion. We have also expanded the explanation for the inclusion of the chosen actors, given their extensive participation in the public sphere during the debate, both in the media and in the consultative commissions that took place in the legislature.
We include note 2 in the text, expanding the explanation of the complexity of the evangelical universe.
With regard to the section "Materials and Methods", we have changed its position in the text, since it followed the article model proposed by the Journal. The material in this section is part of the analysis in the "Results and Discussion" section, although it was not always made explicit, so we clarified it. It is on the basis of this work that we selected the religious groups and actors that were interviewed, as a sample of the complexity of the field.

Reviewer 3 Report

I am sending my comments in relation to a good text but which requires some clarifications to be published.

1. The terms "conservative" and "progressive" are usually not very clarifying categories and belong more to the social or political ideological debate, which is usually the one who defines them. In order to use these terms, it is necessary to explain, even if only in brief, their meanings or the meaning given to them by the author.

2. The following phrase: "One indicator of the consolidation of democracy in Argentina is the expansion of rights, in particular, those concerning the acceptance of sexual and gender diversity and the recognition of women’s autonomy over their body" (18-20)

In some way questions the findings indicated in the conclusions. If, within the framework of democracy, the parliament had not voted such laws, would democracy in Argentina not be consolidated?

What consolidates democracy if not the approval or disapproval of proposed rules?  Perhaps the statement suggests the author's position on the subject, but it does not seem pertinent in academic terms.

3. There are two statements made that I understand need references or clarification for their substantiation:

- "The diversification of the conservative religious field (which in Argentina originally included only members of Catholicism" (29)

- "Their regularity exhibits them as a consolidated and global modus operandi". (470-471)

Both phrases appear to be taxative without reference to studies or sources to support them.

4. I find the following statement inadequate because of its use of two concepts: "Is the public intervention of religious agents a reason to be suspicious, or should their intention to provide public reasons/justifications understandable to all sociopolitical agents be recognized as authentic?" (474-476)

Is the concept "suspicious" adequate? Suspicious of what? Likewise, the concept "authentic" is clearly inadequate, since who can say whether something is "authentic" from a social science point of view. What is socially valued as authentic, true or accepted is a historically situated cultural production. Therefore, I suggest rephrasing the phrase

Author Response

Thank you very much for your suggestions and comments, they were central to the revision of the article.
1.    We explained the use of the term "conservative" for the purposes of the article, adopting a broad definition that refers to those actors who oppose abortion, regardless of whether they are conservative in other fields, or do not share positions with other groups outside the topic of interest. We eliminated "progressive" since it had only been referred to once.
2.    We corrected this phrase, as we agree that it positioned us. We replaced it with the following "An indicator of the consolidation of democracy in Argentina is the openness to diversity, the consolidation of the right to freedom of expression, faith, and autonomy over one's own life and body. One form of this openness is the possibility of discussion and expansion of so-called sexual and reproductive rights".
3.    We include quotations that clarify the phrases indicated. In the first case, texts by Campos Machado and Bárcenas Barajas, wake clear how the conservative camp has widened, adding evangelical activism to the traditional Catholic opposition to the enactment of laws such as the legalization of abortion, egalitarian marriage, etc. In the second case, taking up a previously cited text by Juan Marco Vaggione, where the author explains how a series of strategies of conservative activism (citizenship of the fetus, politicization of the figure of conscientious objection, criminalization of women who have abortions) are replicated in different countries almost simultaneously, showing the aforementioned homogeneous and consolidated modus operandi.
4.    We agree with the inadequacy of the phrase, we replace it with the following: "Is the public intervention of religious actors a problem for diversity in democracy, as it would imply an imposition of their own beliefs as those to be followed by the whole society, or is their intervention a reasonable part of the public debate, a genuine intention to offer public reasons/justifications understandable to all socio-political actors?"

Round 2

Reviewer 1 Report

I'm not sure if I was sent the wrong version of the manuscript, since I see very few changes, mostly in the introductory sections. In my previous report, I had given a 'major' because, although the paper is promising, it still needs some work. Since the authors haven't properly addressed any of my remarks (with the exception of the position of the methodological paragraph), my evaluation is still the same.

Author Response

Thank you for your comments. We have clarified our understanding of the democratic debate in the context of the case analyzed in lines 48-57 and 555-559 of this latest version of the paper.
Regarding the discussion of Peronism, we agree that it is very interesting, but it was beyond the scope of this article. However, we clarify in lines 79-83 that we are only considering political participation in the abortion debate, without delving into party political traditions. We hope to explore this aspect further in future research.
We are trying to establish a language that is less oriented to one of the positions in the conflict. Comments on methodology and interviews were resolved in the previous round.
Thank you very much for your suggestions, which allowed us to discuss the central aspects of the article.

Reviewer 2 Report

Footnote 4 (2 in original version) appears duplicated in p2

I still don´t see the value of the data mentioned in ls. 109-119; I suggest either to A) spell out the value of those data for the author/s' ensuing discussion, B) include and discuss the data in section 3, or C) remove the data.

Author Response

Thank you very much for the new comments. 
We have removed the duplication with the footnote.
Regarding the comment about the social network data, we elaborate on its usefulness for our analysis in lines 136-144 and 159-162 of this latest version. The comment allowed us to detail how the social network analysis was a contribution to reconstructing the different spaces in which the abortion debate unfolded and their characteristics.

Round 3

Reviewer 1 Report

The manuscript is now publishable.